# Puncturing apple fruits increases survival of *Grapholita molesta* (Lepidoptera: Tortricidae) in laboratory rearing

Souvic Sarker[1], Un Taek Lim[2]*

1 Department of Entomology, Rutgers University, New Brunswick, NJ, United States of America,
2 Department of Plant Medicals, Andong National University, Andong, Republic of Korea

* utlim@andong.ac.kr

## Abstract

*Grapholita molesta* (Busck) is a major pest in orchards of apple, peach, and plum. For better rearing in the laboratory, we compared the life history characteristics of *G. molesta* by providing larvae with either punctured or unpunctured apple fruits. The development time of immatures and the fecundity of adult females were similar between punctured and unpunctured apples. However, the overall survival rate of *G. molesta* (larvae to adult emergence) was 1.7 times higher on punctured apples than unpunctured ones, resulting in a higher intrinsic rate of population increase. Therefore, punctured apples would be a better food source for rearing of *G. molesta*.

## Introduction

Oriental fruit moth, *Grapholita molesta* (Busck) (Lepidoptera: Tortricidae), is a major pest of important tree fruits, including peaches, apples, and pears, belong to Rosaceae family [1–5]. *Grapholita molesta* is widely distributed throughout Asia, Europe, the Americas, Africa, and Australia's temperate and subtropical regions [1,4]. It is also a quarantine pest that affects shipment of fruit between countries [6]. Depending on the temperature and location, the pest can have three to six generations per year [7–11].

According to a survey (1992–2005) from Korea, fruit damage by *G. molesta* at harvest varied from 0.02 to 1.64%, and the orchard infested with *G. molesta* was 13 to 71% [12]. In some Brazilian orchards (in 1985), *G. molesta* was reported to have damaged up to 90% of all apples [13]. Since its introduction into the United States in the early twentieth century, *G. molesta* has become a serious pest of tree fruits in that country [14].

Organophosphate, carbamate, and synthetic pyrethroid pesticides are commonly used to control *G. molesta* [15–18], but insecticide resistance poses a serious threat to the fruit industry [17], and *G. molesta* has developed resistance to 14 insecticides, including 10 organophosphates [19]. To address this problem, new IPM programs must be developed and implemented. Preliminary trials in Bulgaria have shown that population densities of *G. molesta* on peaches were reduced during a pilot program that combined classic sterile insect release (SIR) with F1 male sterility [20]. *Grapholita molesta* can also be potentially controlled by using natural enemies [21,22].

**Data Availability Statement:** All relevant data are within the paper and its Supporting Information files.

**Funding:** This work was supported by Korea Institute of Planning and Evaluation for Technology

in Food, Agriculture, and Forestry (IPET) through Agricultural Machinery/Equipment Localization Technology Development Program, funded by Ministry of Agriculture, Food, and Rural Affairs (MAFRA) (321054-05-2-HD020). The funders had no role in study design, data collection and analysis, decision to publish, or preparation of the manuscript.

**Competing interests:** The authors have declared that no competing interests exist.

To provide insects for bioassay and for inexpensive production of natural enemies, an efficient diet and rearing system for *G. molesta* is needed. Green immature apples have been shown to be a good food source for *G. molesta* larvae [23,24]. To rear *G. molesta*, Vetter et al. [25] compared an artificial diet to a punctured-apple diet and found that the punctured-apple diet produced more pupae than the artificial diet. However, many important details, such as the number of punctures, larval development period, fecundity, and female longevity were not reported in that study. Also, no comparison of insect survival has been made of punctured versus unpunctured apples as rearing diets for *G. molesta*. Therefore, in this study, we hypothesized that punctured apple fruit may increase fitness of *G. molesta* by enhancing development and reproduction. We compared biological and life table parameters of *G. molesta* reared on between punctured and non-punctured apple fruits.

## Materials and methods

### Insect rearing

Larvae were reared on apple fruits described by Sarker and Lim [26]. Naturally infested apples were collected from Bioresource Research Institute (Andong, Republic of Korea) in 2015 and kept in ventilated plastic containers (24.0 L × 17.0 W × 8.0 H cm) in a growth chamber (DS-11BPL, Dasol Scientific Co. Ltd, Hwaseong, Republic of Korea) held at 24.9 ± 0.1°C, 50.2 ± 1.3% RH, and a 16:8 h (L:D) photoperiod. After about 20 d of collection, larvae reached the fifth instar and emerged from apples to build their cocoons. To support pupation, a paper towel was placed in boxes. Pupae were then collected about 10 d and held in different breeding dishes (10.0 D × 4.0 H cm, 310102, SPL, Pocheon, Republic of Korea) for adult emergence. After adult moths emerged, 10–15 pairs of adult moths were transferred into ventilated acrylic oviposition cylinders (25.5 H × 8.5 D cm), and a piece of cotton soaked with a 10% sugar solution was provided as an adult food source in each cage. These acrylic cylinders were kept at 25.6 ± 0.1°C and 91.2 ± 0.1% RH in a growth chamber. These oviposition cylinders were examined and changed daily to collect 1-day-old eggs starting when moths first began to lay eggs on the wall. The acrylic cylinders with eggs on the walls were kept in a separate growth chamber at 25.1 ± 1.5°C and 94.3 ± 5.4% RH until the eggs hatched, after which the first instar larvae were collected for use in experiments, or were returned to the mass rearing. OFM was reared for about six generations while experiments were carried out. Wild males were field collected and added to the mass rearing 2–3 times in a year to reduce inbreeding depression.

### Plant materials

Pruned fruits of green fresh apple [*Malus domestica* Borkh. Variety "Fuji" (strain 'Busa')] as an experimental unit were collected from unsprayed apple orchards in Gilan County, Andong City, the Republic of Korea, in 2017. Apples were measured to be 6.2 ± 0.2 cm diameters and sealed in plastic zipper bags held at 4°C in a refrigerator before being used in our experiment.

### Development and survival rate of immature stages

In this experiment, there were two treatments: punctured and unpunctured apple fruits. Punctures were made (2.6–3.2 mm deep) with an insect pin (Insect Pins, Stainless steel No. 6, Bio-Quip Products, California, USA), with 20 punctures scattered around the apple fruit. Before being used, insect pins were sterilized with an alcohol lamp, followed by dipping in 70% ethanol. After preparation, punctured and unpunctured apples were placed individually in insect breeding dishes (310122, 120 D × 80 H mm, SPL, Pocheon, Republic of Korea). Five larvae (<5 h old) of *G. molesta* were then placed on the surface of each fruit with the help of camel

hair brush, and fruits were then held at 24.9 ± 0.1˚C and 51.7 ± 1.6% RH in a growth chamber. In total, in this experiment, there were 30 larvae used in each of the punctured and unpunctured apple treatments with six replications (5 insects/ apple fruit). When a mature larva emerged from a fruit (exiting larva), it was transferred for pupation into another dish (10.0 D × 4.0 H) that was filled with tissue paper. The duration of the larval stage was estimated as the time from the day of egg hatch (= the day fruit were inoculated with neonate larvae) to the day the mature larva exited the fruit. The duration of the prepupal stage was the period from the exit of the mature larva from the fruit to the pupation, and the pupal stage was from the day of pupation to adult emergence.

## Longevity and fecundity of adult females

Adults of *G. molesta* reared from larvae in treatments were transferred into transparent square breeding dishes (7.2 L × 7.2 W × 10.0 H cm SPL, Pocheon, Republic of Korea, each with three 40 mm-mesh screens; one per side) immediately after their emergence and held with in pairs (1M, 1F) per dish. A piece of cotton soaked with a 10% sugar solution was added to each cage as a food source. Freshly laid eggs were deposited on the container's surface, and eggs were marked and recorded daily until the adult female died. Breeding dishes holding adult pairs were replaced daily with new ones to avoid pathogenic infections. An absence of spermatophores was used to indicate an unmated female, and such moths were excluded from the analysis [5] and a total of 11 females for punctured and 6 for unpunctured were used in this study. If the male died before the female or if there was uneven number of male and female produced from the treatment, a new male was provided from the cohort cage.

## Statistical analysis

Differences in developmental time, preoviposition period, oviposition period, lifetime fecundity, the longevity of adult females, and pupal weight of *G. molesta* for animals reared on punctured versus unpunctured apple fruits were analyzed with *t*-tests using SAS 9.4 [27]. Survival rate of each stage was analyzed using two proportion *Z*- tests [28].

The jackknife procedure was performed to test the differences in population parameters, i.e., doubling time ($DT$), finite rate of increase ($\lambda$), intrinsic rate of increase ($r_m$), net production rate ($R_O$), and mean generation time of ($T$) [29]. The survival rate ($S_{xj}$) ($x$ = age, $j$ = stage) is the probability that a newly laid egg would survive to age $x$ and stage $j$. Jackknife algorithms for estimating the means and variances and constructing confidence intervals were described only for $R_O$ (the net contribution of each female to the next generation, expressed as the total female offspring per female over the whole oviposition period) [29]. The same procedures were used to estimate other parameters ($r_m$, $T$, $DT$, and $\lambda$). All fertility data in the life tables were entered in a computer program (LIFETABLE.SAS) [29] and analyzed using SAS 9.4 [27].

## Results

### Development and survival rates of immature stages

No significant differences were found in durations of egg stage ($t$ = 0.27, *df* = 58, $P$ = 0.791), larval stage ($t$ = 0.13, *df* = 33, $P$ = 0.898), prepupal stage ($t$ = 0.57, *df* = 28, $P$ = 0.576), pupal stage ($t$ = 1.84, *df* = 28, $P$ = 0.076), or immature stage ($t$ = 1.36, *df* = 28, $P$ = 0.183) between punctured and unpunctured apple diets (Table 1).

The rate larvae exited from fruits ($Z_c$ = 1.83, $P$ = 0.067), the pupation rate ($Z_c$ = 0.99, $P$ = 0.324), and the adult emergence rate ($Z_c$ = 0.00, $P$ = 1.000) were also not different between punctured and unpunctured apples, but the overall survival rate ($Z_c$ = 2.07, $P$ = 0.039) was

**Table 1. Duration (d) (± SE) of each stage of *Grapholita molesta* (n = 30 for punctured and n = 30 for unpunctured) reared on apple fruits under laboratory conditions.**

| Treatment | Egg | Larva | Prepupa | Pupa | Immature |
|---|---|---|---|---|---|
| Punctured | 3.37 ± 0.09 a (n = 30) | 15.52 ± 0.48 a (n = 21) | 3.95 ± 0.38 a (n = 19) | 8.95 ± 0.40 a (n = 19) | 28.58 ± 0.54 a (n = 19) |
| Unpunctured | 3.33 ± 0.09 a (n = 30) | 15.43 ± 0.54 a (n = 14) | 4.36 ± 0.72 a (n = 11) | 10.36 ± 0.74 a (n = 11) | 30.27 ± 1.35 a (n = 11) |

Means within a column with different letters are significantly different ($P < 0.05$).

significantly different between punctured and unpunctured apples (Table 2). The overall survival rate of *G. molesta* larvae in punctured apples was 60%, which was higher than that of larvae reared in unpunctured apples (40%).

The survival rates of *G. molesta* larvae reared on punctured and unpunctured apples were highest on 23rd and 21st day, respectively (Fig 1). We found that no fungal development occurred within three weeks.

## Longevity and fecundity of adult females

No significant differences were found in the preoviposition period ($t = 0.38$, $df = 5$, $P = 0.721$), oviposition period ($t = 0.49$, $df = 5$, $P = 0.643$), lifetime fecundity ($t = 0.22$, $df = 5$, $P = 0.833$), or adult female longevity ($t = 0.45$, $df = 5$, $P = 0.671$) between insects reared on punctured versus unpunctured apples (Table 3). Pupal weights ($t = 1.36$, $df = 28$, $P = 0.183$) were also similar between treatments (Fig 2).

## Life history parameters of *Grapholita molesta*

Significant differences were observed between punctured and unpunctured apple in several life table parameters, including doubling time ($DT$), finite rate of increase ($\lambda$), intrinsic rate of increase ($r_m$), and net reproductive rate ($R_O$) (Table 4). Intrinsic rate of increase ($r_m$) was significantly higher on punctured apples, and the doubling time was significantly shorter (i.e., faster population growth) on punctured apples than in moths reared on unpunctured apples. However, mean generation time ($T$) was similar between punctured and unpunctured apples (Table 4).

## Discussion

*Grapholita molesta* has been mass-reared in artificial diets for many years [5,25,30,31]. Green immature apples have also been used for rearing larvae of *G. molesta* and are a good food source for the development and reproduction [23,24,32–34]. First and second instars of *G. molesta* are highly cannibalistic when grown on an artificial diet lacking protein and vitamins, while they are not cannibalistic when grown on green apples even under crowded conditions [35]. Furthermore, Löfstedt et al. [36] found that the amount of ethyl *trans*-cinnamate is higher

**Table 2. Survival rate of each stage of *Grapholita molesta* provided with punctured and unpunctured apple fruits under laboratory conditions.**

| Treatment | Exiting rate | Pupation rate | Emergence rate | Overall survival rate |
|---|---|---|---|---|
| Punctured | 0.70 (21/30) a | 0.90 (19/21) a | 1.00 (19/19) a | 0.63 (19/30) a |
| Unpunctured | 0.47 (14/30) a | 0.79 (11/14) a | 1.00 (11/11) a | 0.37 (11/30) b |

Means within a column with different letters are significantly different ($P < 0.05$).

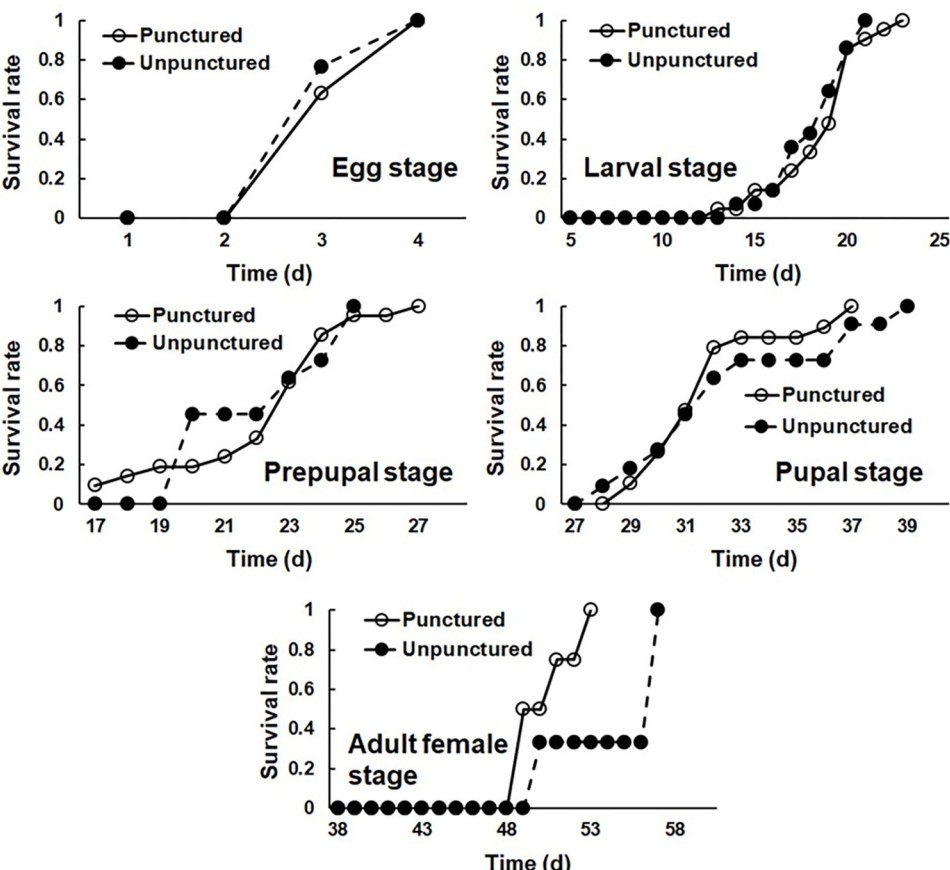

**Fig 1. Survival rate of different life stages of *Grapholita molesta* reared on punctured and unpunctured apples.**

on adult males reared on apple fruits rather than on artificial diet. Ethyl *trans*-cinnamate is an important component of the sex pheromone of *G. molesta* [37], and *G. molesta* males sequester compounds from apples to produce ethyl *trans*-cinnamate for close-range attraction of calling females [37,38]. In addition, males reared in apple fruits can attract females from a longer distance than those reared on artificial diet [36].

Vetter et al. [25] compared an artificial diet with the use of punctured apples for mass rearing of *G. molesta* and found that the punctured apple diet yielded more pupae than the artificial diet. However, this study did not indicate the number of punctures, the larval developmental period, or adult fecundity or female longevity. Furthermore, Vetter et al. [25] made no comparison for *G. molesta* survival in apples with punctures versus unpunctured apples.

In our study, the durations of the egg, larval, prepupal, pupal, and immature stages were not statistically different between punctured and unpunctured apples. The larval

**Table 3. Fecundity (Mean ± SE) of female adult of *Grapholita molesta* reared on punctured and unpunctured apple fruits under laboratory conditions.**

| Treatment | Preoviposition period | Oviposition period | Lifetime fecundity | Longevity |
|---|---|---|---|---|
| Punctured | 3.50 ± 0.29 a | 17.30 ± 1.31 a | 238.50 ± 8.97 a | 20.80 ± 1.25 a |
| Unpunctured | 3.33 ± 0.33 a | 18.33 ± 1.86 a | 234.33 ± 18.41 a | 21.67 ± 1.67 a |

Means within a column with different letters are significantly different ($P < 0.05$).

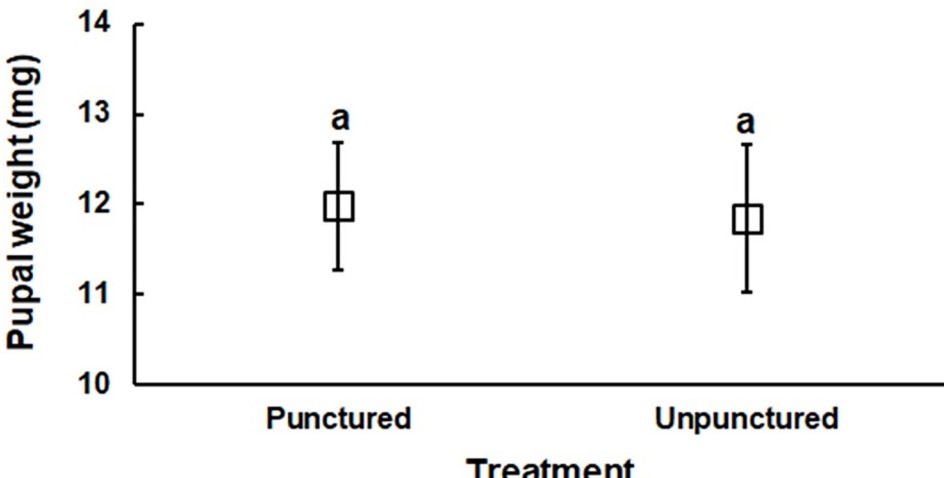

**Fig 2. Pupal weights of *Grapholita molesta* reared on punctured and unpunctured apples.**

developmental times on punctured and unpunctured apple were 15.5 and 15.4 days, respectively. According to Bisognin et al. [30], the larval period of *G. molesta* reared on apple (cv. Fuji) is 15.9 days, while it is shorter on artificial diet. Generally, the larval developmental time was longer in apple (the preferred host) than in peach or plum fruits [23,24].

Also, there were no significant differences in the rate of fruit exiting for larvae, the pupation rate, or the emergence rate between punctured and unpunctured apples. However, the overall survival rate was higher in punctured apples (60%). When rearing *G. molesta* larvae on apple fruits at the rate of two larvae per apple, Bisognin et al. [30] found that the total survival rate of *G. molesta* was only 29.9%. This is much lower than the 54.8% for artificial diet. Here, we found a similarly low survival rate (36.7%) in unpunctured apples with five larvae per apple, but in punctured apples survival was much higher (60%). The lower survival rates in unpunctured apples are most likely caused by higher number of larvae being unable to enter the fruit; punctures seem necessary to avoid this mortality and maximize *G. molesta* production efficiency. In other cases, Finney et al. [37] found that larvae of *Phthorimaea operculella* (Zeller) could enter the potato more effectively if the skin was punctured, which permitted large numbers of *G. operculella* to be reared in individual potatoes. Unpunctured potatoes are infested only by larvae that enter through the potato eyes, reducing the number of larvae that can be reared from each tuber [39].

**Table 4. Life *table parameters of G. molesta reared on punctured and unpunctured apples.***

| Treatment | Parameters (mean ± SE) | | | | |
|---|---|---|---|---|---|
| | *DT* | $\lambda$ | $r_m$ | $R_o$ | *T* (Day) |
| Punctured | 5.02 ± 0.21 b | 1.15 ± 0.01 a | 0.14 ± 0.01 a | 87.15 ± 3.28 a | 32.34 ± 1.15 a |
| Unpunctured | 5.96 ± 0.13 a | 1.12 ± 0.00 b | 0.12 ± 0.00 b | 47.69 ± 3.75 b | 33.25 ± 0.62 a |

Means followed by the same letter in a column are not significantly different from Student's *t*-test for pairwise group comparison at $P < 0.05$.

*DT*: Doubling time

$\lambda$: Finite rate of increase

$r_m$: Intrinsic rate of increase

$R_O$: Net reproductive rate

*T*: Mean generation time.

We found no significant differences for *G. molesta* between punctured and unpunctured apples in pupal weight, the preoviposition period, oviposition duration, lifetime fecundity, and longevity. This suggests that larvae that do successfully enter apple fruits obtain the necessary nutrients. Because of differences in overall survival rate, the intrinsic rate of increase was higher in *G. molesta* reared in punctured apples than those reared in unpunctured apples. This difference is probably due to the higher rate of the first instar larvae successfully enter the fruit (Table 1), resulting in higher overall survival rate. However, in field condition, physical damage on apple can negatively influence the population dynamics of *G. molesta* by plant volatiles produced to attract natural enemies [40,41].

Larvae of *G. molesta* prefer green, immature apples as a food source and normally enter the apple fruit through the calyx or stem end cavities [42]. Several researchers have reported a decline over time in the quality of *G. molesta* reared in ripe apples because ripe apples decline quickly in quality [35,43–45]. Similarly, in our previous study, ripe plum fruit was not a good food source for *G. molesta* larvae due to the decline in fruit firmness [24]. Therefore, *G. molesta* can be simply and cheaply reared using cold-stored, green, immature apples even though obtaining large quantities of green apples, and storing them without loss of quality, can be difficult. Most importantly, we show that the efficiency of rearing OFM in green apples can be greatly increased by puncturing green immature apples to allow for better ingress of neonate larvae.

## Supporting information

**S1 File. Punctured apple, unpunctured apple, life table analysis, pupal weight.**
(XLSX)

**S1 Appendix. Demonstrating the puncture to the apple fruit.**
(TIF)

## Author Contributions

**Conceptualization:** Souvic Sarker, Un Taek Lim.

**Data curation:** Souvic Sarker, Un Taek Lim.

**Formal analysis:** Souvic Sarker, Un Taek Lim.

**Funding acquisition:** Un Taek Lim.

**Investigation:** Souvic Sarker, Un Taek Lim.

**Methodology:** Souvic Sarker, Un Taek Lim.

**Resources:** Un Taek Lim.

**Supervision:** Souvic Sarker, Un Taek Lim.

**Validation:** Souvic Sarker, Un Taek Lim.

**Visualization:** Souvic Sarker, Un Taek Lim.

**Writing – original draft:** Souvic Sarker, Un Taek Lim.

**Writing – review & editing:** Souvic Sarker, Un Taek Lim.

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
