## [Decision Letter · Decision Letter 0]

28 Feb 2022

PONE-D-22-02750Puncturing Apple Fruits Increases Survival of Grapholita molesta (Lepidoptera: Tortricidae) in Laboratory RearingPLOS ONE

Dear Dr. Lim,

Thank you for submitting your manuscript to PLOS ONE. After careful consideration, we feel that it has merit but does not fully meet PLOS ONE’s publication criteria as it currently stands. Therefore, we invite you to submit a revised version of the manuscript that addresses the points raised during the review process.

We look forward to receiving your revised manuscript.

Kind regards,

Patrizia Falabella

Academic Editor

PLOS ONE

Journal Requirements:

"This work was supported by Korea Institute of Planning and Evaluation for Technology in Food, Agriculture, and Forestry (IPET) through Agricultural Machinery/Equipment Localization Technology Development Program, funded by Ministry of Agriculture, Food, and Rural Affairs (MAFRA) (321054-05-2-HD020)."

"This work was supported by Korea Institute of Planning and Evaluation for Technology in Food, Agriculture, and Forestry (IPET) through Agricultural Machinery/Equipment Localization Technology Development Program, funded by Ministry of Agriculture, Food, and Rural Affairs (MAFRA) (321054-05-2-HD020).

Reviewers' comments:

Reviewer's Responses to Questions

**Comments to the Author**

1. Is the manuscript technically sound, and do the data support the conclusions?

Reviewer #1: Yes

Reviewer #2: Partly

Reviewer #3: No

2. Has the statistical analysis been performed appropriately and rigorously? 

Reviewer #1: N/A

Reviewer #2: N/A

Reviewer #3: No

3. Have the authors made all data underlying the findings in their manuscript fully available?

Reviewer #1: Yes

Reviewer #2: Yes

Reviewer #3: Yes

4. Is the manuscript presented in an intelligible fashion and written in standard English?

Reviewer #1: Yes

Reviewer #2: Yes

Reviewer #3: Yes

5. Review Comments to the Author

Reviewer #1: This study showed the needle punctured green (?) apple could be a better substrate for Oriental fruit moth larval rearing medium than not-punctured ones.

The experiments were well designed and conducted.

Writing is concise and without redundancy.

Results were reasonable.

It is difficult to judge the scientific merit of this finding in science but surely important in applied aspects.

M&M Please provide when and how did the field collection of apple was done and detial the condition of apple collected.

Table 1. egg duration was presented. However in M&M section, larvae were introduced into the experimental apples and began observation. This part could be cleared.

In Table 1, provide the actual numbers of observation for each stage.

Exiting rate was measured but not described in M&M. Provide the details of the measurement. In the box, if the larvae exit the fruit then,does this mean that the larvae became dead, on what time scale?

please check the statistics; the rate larvae exited from fruits (Zc = 1.83, P = 0.067), the pupation rate (Zc = 0.99, P = 134 0.324), and the adult emergence rate (Zc = 0.00, P = 0.000) were also not different between punctured and unpunctured apples,

Authors discussed the punctured apple could provide easy approaches to the week larvae to enter into the fruits and lowering larval mortality.

In the study, separation of larval mortality and exiting rate, and pupal mortailty and pupation rate was not explicitly detailed, and which made the overall conclusion less supporting.

In the table showing adult reproduction, please provide the numbers of female observed for oviposition.

In this study, how many pairs (male and female) were studied were not presented. If If there was uneven number of male and female produced from the treatment, did you add the male from the rearing batch or ...

Authors had narrowed the scope of this finding only in laboratory rearing, however this findig could influence field dynamics of infestation.

By puncturing, the host apple could provide more VOCs which would influence the host location to the larvae. Please add the possibility of this in the discussion, relative to the field situation where diverse factors could impact apple fruits and give physical damages.

In this study, can we have any data of apple physical character change over the experimental periods? Since the larvae should stay ap 2 weeks inside the host food once it entered into, physical changes of the host food could influence the outcome of the results.

Reviewer #2: In this work the authors study the effect of punctured apple fruits on survivorship of G.molesta. Based on their analysis the authors conclude that punctured fruits are more suitable for OFM laboratory rearing compared to non-punctured fruits and as it is shown by the demographic parameters that the authors have estimated including the intrinsic rate of increase.

These results support previous works done in G. molesta and are generally interesting and important for OFM rearing. In this respect the authors reaffirm the importance of their work to the fact that their results will contribute to the mass production of this pest to be used for mass rearing. On the other hand, I feel that the current work is rather simplistic and based on a rather small data set.

I cannot imagine also how practical might be the mass production of OFM in a natural host compared to artificial diet despite the better demographic performances of the species. For example, injured apples will not only improve OFM growth (at least as the present work demonstrates) but also the growth of fungal infections and ultimately cannot be used for a long time being impractical. i.e. How many times have you changed the apples during larval development? How easy it is to mechanize the technique you are proposing for OFM mass rearing?

Additionally, the study does not examine whether the ''good reproductive performance'' that the authors have observed (through the two comparisons: punctured vs unpunctured hosts) holds also for other temperature regimes. Because the intrinsic rate of increase takes in to account at least both, the net maternity along with survivorship of the individuals of a cohort, it is quite possible that the differences might vanish under different temperature regimes. In table 4, for instance, the distance in rm between the two treatments is considerable shorter compared to Ro. Actually, I consider the intrinsic rate of increase as more representative for the description of the reproductive success of species (i.e. fitness) compared to the net maternity and not exclude that this difference might diminished after a slight alteration of the temperature regime.

Finally, the authors focus on the technical part of their study for OFM rearing (which is sound and not bad), but, to my view incomplete because they have not examined in deep why they have observed such significant differences. Particularly and if I am not wrong, a part of the last lines (213-214), there is actually no hypothesis of what is the profound reason-interpretation of their result and especially what is(are) the main reason(s) that the species explicit different fitness capacities when feeded with an injured host.

If it’s just the fact that it’s easy (and not energy consuming) for the larvae to crawl insight an injured apple compared to an unpunctred one, how do they explain the differences that are observed on the successive developmental stages? Are they excluding any nutritional, physiological or any other cause?

Thus, I believe that the biological-ethological and consequently the ecological dimension of the authors' findings is of greater interest but not emphasized at all.

Line 45. If this proposition were to the maximum, i.e. biological was effective, we would not need research on pesticides or even deprivation.

Line 58. It would help the reader if you included a photo in the appendix

Figure 1. To be honest I cannot follow your chart. Why do you not just plot the survivorship curves for each stage and treatment?

Reviewer #3: Puncturing Apple Fruits Increases Survival of Grapholita molesta (Lepidoptera: Tortricidae) in Laboratory Rearing By Souvic Sarker and Un Taek Lim

General comments

This is a study where unpunctured and punctured apples were compared as diet for artificially rearing the pest Oriental fruit moth, Grapholita molesta. To determine the effect of these two diets, authors measured and analysed some demographic parameters in the reared insects. The article has some interesting results. However, the results are not novel and, results are poorly presented. The manuscript has a very narrow scope, basically in the field of entomologists. My major concern is the experimental design since there is not explanation of what the experimental unit was, the apple or the insect.

Introduction

Lines 46-47. A mass-rearing system based on apples. It is not possible. Fruit phenology and fruit variation surely will introduce many difficulties in the process.

Methods

Line 67. How many insect per cylinder?

Describe with detail the rearing method. Did the authors place the larva on apple surface? How did they do it?

Line 75. Please indicate the periodicity.

Lines 84-96. Did you use 12 apples as a total experimental set? Six apples per treatment? Please clarify.

Lines 100-109. Were the apples the experimental unit? Or the larvae? Five larvae per fruit gives 30 individual insects, some of them died so how many did the authors use for determining the demographic parameters?

Results

Figures can be deleted and the values presented on them can be included in the text.

6. PLOS authors have the option to publish the peer review history of their article (what does this mean?). If published, this will include your full peer review and any attached files.

Reviewer #1: No

Reviewer #2: No

Reviewer #3: No

---

## [Author Response · Author response to Decision Letter 0]

13 Apr 2022

A. Response to reviewer #1’s comments

1. This study showed the needle punctured green (?) apple could be a better substrate for Oriental fruit moth larval rearing medium than not-punctured ones.

The experiments were well designed and conducted. Writing is concise and without redundancy. Results were reasonable. It is difficult to judge the scientific merit of this finding in science but surely important in applied aspects.

>> Thanks for the comments.

2. M&M Please provide when and how did the field collection of apple was done and detial the condition of apple collected.

>> We provided the time and collection procedure in the revised MS. Please see L80-82 in the revised MS.

3. Table 1. egg duration was presented. However in M&M section, larvae were introduced into the experimental apples and began observation. This part could be cleared.

>> We addressed this issue in our previous MS. See L71-75 in the revised MS.

4. In Table 1, provide the actual numbers of observation for each stage.

>> Yes. Please see Table 1 in revised MS.

5. Exiting rate was measured but not described in M&M. Provide the details of the measurement. 

>> We corrected the sentence by adding “exiting larvae”. Please L95-96 in the revised MS.

6. In the box, if the larvae exit the fruit then, does this mean that the larvae became dead, on what time scale?

>> We addressed this issue in our previous MS in L93-94. When a larva exits from the fruit, it transferred into another dish for pupation. Please see L95-98 in the revised MS.

7. please check the statistics; the rate larvae exited from fruits (Zc = 1.83, P = 0.067), the pupation rate (Zc = 0.99, P = 134 0.324), and the adult emergence rate (Zc = 0.00, P = 0.000) were also not different between punctured and unpunctured apples, Authors discussed the punctured apple could provide easy approaches to the week larvae to enter into the fruits and lowering larval mortality.

>> As we mentioned in our previous MS in L184-185 and L202-204, the overall survival rate and entering rate of the 1st instar was higher in punctured apples despite the lack of significance in each parameter, i.e., exit rate of larvae, pupation rate, adult emergence rate. Please see L185-186 and L203-204 in the revised MS.

Nevertheless, we deleted Figure 2 as it is redundant data of Table 2. 

8. In the study, separation of larval mortality and exiting rate, and pupal mortailty and pupation rate was not explicitly detailed, and which made the overall conclusion less supporting.

>> Please see our answer to above comment.

9. In the table showing adult reproduction, please provide the numbers of female observed for oviposition.

>> Yes. Please see L110-111 in the revised MS.

10. In this study, how many pairs (male and female) were studied were not presented. If If there was uneven number of male and female produced from the treatment, did you add the male from the rearing batch or ...

>> We addressed this issue in L110-113 of the revised MS.

11. Authors had narrowed the scope of this finding only in laboratory rearing, however this findig could influence field dynamics of infestation.

>> We agree with the reviewer’s comment. We added “However, in field condition, physical damage on apple can negatively influence the population dynamics of G. molesta by plant volatiles produced to attract natural enemies” in the revised MS. Please see L205-207 in the revised MS.

12. By puncturing, the host apple could provide more VOCs which would influence the host location to the larvae. Please add the possibility of this in the discussion, relative to the field situation where diverse factors could impact apple fruits and give physical damages.

>> Please see our answer to above comment

13. In this study, can we have any data of apple physical character change over the experimental periods? Since the larvae should stay ap 2 weeks inside the host food once it entered into, physical changes of the host food could influence the outcome of the results.

>> We addressed this issue in our previous MS in L206-208. Please see L209-211 in the revised MS. 

B. Response to reviewer #2’s comments

1. In this work the authors study the effect of punctured apple fruits on survivorship of G.molesta. Based on their analysis the authors conclude that punctured fruits are more suitable for OFM laboratory rearing compared to non-punctured fruits and as it is shown by the demographic parameters that the authors have estimated including the intrinsic rate of increase.

These results support previous works done in G. molesta and are generally interesting and important for OFM rearing. In this respect the authors reaffirm the importance of their work to the fact that their results will contribute to the mass production of this pest to be used for mass rearing. On the other hand, I feel that the current work is rather simplistic and based on a rather small data set.

1. I cannot imagine also how practical might be the mass production of OFM in a natural host compared to artificial diet despite the better demographic performances of the species. For example, injured apples will not only improve OFM growth (at least as the present work demonstrates) but also the growth of fungal infections and ultimately cannot be used for a long time being impractical. i.e. How many times have you changed the apples during larval development? How easy it is to mechanize the technique you are proposing for OFM mass rearing?

>> We agree with the reviewer’s concern that this rearing method using natural host can’t be appropriate for commercial mass-production. This is why we just suggested that punctured apple is better than unpunctured apple throughout MS although Vetter et al. [25] found that the punctured-apple diet produced more pupae than the artificial diet. 

Nevertheless, we added a sentence in result section regarding fungal contamination on punctured apple. See L143-144 in the revised MS.

2. Additionally, the study does not examine whether the ''good reproductive performance'' that the authors have observed (through the two comparisons: punctured vs unpunctured hosts) holds also for other temperature regimes. Because the intrinsic rate of increase takes in to account at least both, the net maternity along with survivorship of the individuals of a cohort, it is quite possible that the differences might vanish under different temperature regimes. In table 4, for instance, the distance in rm between the two treatments is considerable shorter compared to Ro. Actually, I consider the intrinsic rate of increase as more representative for the description of the reproductive success of species (i.e. fitness) compared to the net maternity and not exclude that this difference might diminished after a slight alteration of the temperature regime.

>> We agree with the comments of the reviewer. Different temperature regimes may have an impact on G. molesta's development as well as its intrinsic rate of increase. However, our goal was to see whether puncturing apple fruit might improve the intrinsic rate of increase or not. That is why we maintained the same environmental and fruit conditions. 

3. Finally, the authors focus on the technical part of their study for OFM rearing (which is sound and not bad), but, to my view incomplete because they have not examined in deep why they have observed such significant differences. Particularly and if I am not wrong, a part of the last lines (213-214), there is actually no hypothesis of what is the profound reason-interpretation of their result and especially what is(are) the main reason(s) that the species explicit different fitness capacities when feeded with an injured host.

>> We changed “Therefore, in this study our goal was to compare the effects of punctured versus non-puncture apple fruit on the development, reproduction, and life table parameters of G. molesta.” to “Therefore, in this study, we hypothesized that punctured apple fruit may increase fitness of G. molesta by enhancing development and reproduction. We compared biological and life table parameters of G. molesta reared on between punctured and non-punctured apple fruits.” Please see L54-57 in the revised MS.

4. If it’s just the fact that it’s easy (and not energy consuming) for the larvae to crawl insight an injured apple compared to an unpunctred one, how do they explain the differences that are observed on the successive developmental stages? Are they excluding any nutritional, physiological or any other cause? Thus, I believe that the biological-ethological and consequently the ecological dimension of the authors' findings is of greater interest but not emphasized at all.

>> We thought punctured apple should be artefact thus it may not easily applicable to natural interpretation. Anyway, we added a discussion on impact on field population as suggested by reviewer 1. See L205-207 in the revised MS.

5. Line 45. If this proposition were to the maximum, i.e. biological was effective, we would not need research on pesticides or even deprivation.

>> We corrected the sentence by adding “potentially” in the revised MS. Please see L45 in the revised MS.

6. Line 58. It would help the reader if you included a photo in the appendix

>> We added a photo in the appendix demonstrating the puncture to the apple fruit. Please see L372-374 in the revised MS.

7. Figure 1. To be honest I cannot follow your chart. Why do you not just plot the survivorship curves for each stage and treatment?

>> Yes, we revised the figure and caption. Please see Figure 1 and L363-364 in the revised MS. 

C. Response to reviewer #3’s comments

General comments

1. This is a study where unpunctured and punctured apples were compared as diet for artificially rearing the pest Oriental fruit moth, Grapholita molesta. To determine the effect of these two diets, authors measured and analysed some demographic parameters in the reared insects. The article has some interesting results. However, the results are not novel and, results are poorly presented. The manuscript has a very narrow scope, basically in the field of entomologists. My major concern is the experimental design since there is not explanation of what the experimental unit was, the apple or the insect.

>> Apple fruits were the experimental unit and we revised the sentence. See L81 in the revised MS. 

Introduction

2. Lines 46-47. A mass-rearing system based on apples. It is not possible. Fruit phenology and fruit variation surely will introduce many difficulties in the process.

>> We agree with the reviewer’s concern that this rearing method using natural host can’t be appropriate for commercial mass-production. This is why we just suggested that punctured apple is better than unpunctured apple throughout MS although Vetter et al. [25] found that the punctured-apple diet produced more pupae than the artificial diet. Anyway, we deleted “mass” from the revised MS. Please see L48 in the revised MS.

Methods

3. Line 67. How many insect per cylinder?

>> We deleted “they” and added “10-15 pairs of adult moths” in the revised MS. Please see L68 in the revised MS.

4. Describe with detail the rearing method. Did the authors place the larva on apple surface? How did they do it?

>> We corrected a sentence in L92 in the revised MS. 

5. Line 75. Please indicate the periodicity.

>> We changed “periodically” to “2-3 times in a year”. Please see L77 in the revised MS.

6. Lines 84-96. Did you use 12 apples as a total experimental set? Six apples per treatment? Please clarify.

>> Yes. We added “with six replications (5 insects/ apple fruit).” Please see L95 in the revised MS.

7. Lines 100-109. Were the apples the experimental unit? Or the larvae?

>> Apple fruits were the experimental unit and we revised the sentence. See L81 in the revised MS. 

8. Five larvae per fruit gives 30 individual insects, some of them died so how many did the authors use for determining the demographic parameters?

>> We added the information in L110-113 in the revised MS.

Results

9. Figures can be deleted and the values presented on them can be included in the text.

>> We deleted Figure 2 as it is redundant data of Table 2. 

D. Author’s corrections (Line numbers are from the first MS)

L114: We added “rate” after “survival” in L118 in the revised MS.

L115: We deleted “the percentages of larval and pupal mortality” in L119 in the revised MS.

L115: We changed “were” to “was” in L119 in the revised MS.

L134: We corrected “0.000” to “1.000” in L137 in the revised MS. It was type error. 

L139-140: We changed “Age- and stage-specific survival rates of G. molesta larvae were highest on 20th day in both punctured and unpunctured apple fruits (Fig. 1)” to “The survival rates of G. molesta larvae reared on punctured and unpunctured apples were highest on 23rd and 21st day, respectively (Fig. 1)”. See L142-143 in the revised MS.

L150: We changed “Fig. 3” to “Fig. 2” in L151 in the revised MS.

L186: We deleted “because the larval and pupal mortality rates were lower in punctured fruits” in L187 in the revised MS.

L201: We changed “larval and pupal mortality” to “overall survival rate” in L202 in the revised MS.

L203: We added “(Table 1)” in L204 in the revised MS.

L204: We changed “larval” to “overall” in L205 in the revised MS.

L226: Changed “wprs” to “WPRS” in L229 in the revised MS.

L239: Changed “Sinica” to “Sin” in L242 in the revised MS.

L245 and L266: Changed “Manage” to “Manag” in L248 and L269 in the revised MS.

L312: Deleted “and”. See L315 in the revised MS.

L321: Changed “Chern” to “Chem” in L324 in the revised MS.

---

## [Decision Letter · Decision Letter 1]

19 Apr 2022

Puncturing Apple Fruits Increases Survival of Grapholita molesta (Lepidoptera: Tortricidae) in Laboratory Rearing

PONE-D-22-02750R1

Dear Dr. Un Taek Lim,

We’re pleased to inform you that your manuscript has been judged scientifically suitable for publication and will be formally accepted for publication once it meets all outstanding technical requirements.

Kind regards,

Patrizia Falabella

Academic Editor

PLOS ONE

Additional Editor Comments (optional):

Reviewers' comments:

Reviewer's Responses to Questions

**Comments to the Author**

1. If the authors have adequately addressed your comments raised in a previous round of review and you feel that this manuscript is now acceptable for publication, you may indicate that here to bypass the “Comments to the Author” section, enter your conflict of interest statement in the “Confidential to Editor” section, and submit your "Accept" recommendation.

Reviewer #1: (No Response)

Reviewer #2: All comments have been addressed

2. Is the manuscript technically sound, and do the data support the conclusions?

Reviewer #1: Yes

Reviewer #2: Partly

3. Has the statistical analysis been performed appropriately and rigorously? 

Reviewer #1: Yes

Reviewer #2: Yes

4. Have the authors made all data underlying the findings in their manuscript fully available?

Reviewer #1: Yes

Reviewer #2: Yes

5. Is the manuscript presented in an intelligible fashion and written in standard English?

Reviewer #1: Yes

Reviewer #2: Yes

6. Review Comments to the Author

Reviewer #1: (No Response)

Reviewer #2: The authors have revised the article according to the comments made. In general and apart of not justifiing the relatively small sample size as indicated in my general coment, all other concerns have been more or less addressed. They could also have uploaded the article with trackchanges to fecilitate review.

7. PLOS authors have the option to publish the peer review history of their article (what does this mean?). If published, this will include your full peer review and any attached files.

Reviewer #1: No

Reviewer #2: No

---

## [Editor Report · Acceptance letter]

22 Apr 2022

PONE-D-22-02750R1 

Puncturing Apple Fruits Increases Survival of *Grapholita molesta* (Lepidoptera: Tortricidae) in Laboratory Rearing 

Dear Dr. Lim:

I'm pleased to inform you that your manuscript has been deemed suitable for publication in PLOS ONE. Congratulations! Your manuscript is now with our production department. 

Kind regards, 

on behalf of

Prof. Patrizia Falabella 

Academic Editor

PLOS ONE